# Telemonitoring starting in the emergency department as an alternative to acute hospital admission: A prospective pilot study focusing on patient preferences and first experience

Noortje Zelis[1,2], Dewa Westerman[1], Anouk Schevers[3], Nicole v Eldik[3,4], Patricia M. Stassen[1,2]*

1 Department of Internal Medicine, Division General Medicine, Section Acute Medicine, Maastricht University Medical Center, Maastricht, The Netherlands, 2 Cardiovascular Research Institute Maastricht (CARIM), Maastricht University, Maastricht, The Netherlands, 3 Healthcare Innovation Lab, Maastricht University Medical Center, Maastricht, The Netherlands, 4 Department of Pediatrics, Maastricht University Medical Center, Maastricht, The Netherlands

* p.stassen@mumc.nl

## Abstract

Telemonitoring at home may be used to reduce acute hospital admissions via the emergency department (ED), but experience in this setting is scarce. We performed a pilot study to investigate the perspectives and experiences of ED patients and care professionals with telemonitoring, started in the ED and used as potential an alternative to acute hospital admission. In this prospective pilot study, we asked medical ED patients for their perspectives on home monitoring. Suitability for homemonitoring was assessed by ED patients and care professionals. In a subset of patients, we started and evaluated telemonitoring. In total, 98 patients answered a questionnaire. The facilitators for telemonitoring as an alternative to hospital admission were: guaranteed admission if necessary (indicated by 96.9% of patients), possibility to contact the treatment team 24/7 (by 90.8%), and presence of someone to watch over the patient (by 72.4%). Main barriers for telemonitoring as an alternative care form were: need for treatment that could not be provided at home, feeling too severely ill, and judging it unsafe to return home. In total, 11.2% of ED patients indicated that hospital admission could be avoided using telemonitoring, while another 6.1% thought this might be possible. Professionals judged fewer patients capable of being sent home with telemonitoring (physicians: 7.2% and 6.1%, resp.; nurses: 10.4% and 4.2%, resp.). Agreement on the capability of patients to be sent home with telemonitoring between patients and professionals was slight-fair. All telemonitored patients were satisfied with the ease of use and comfort of the system, which gave most patients reassurance and was considered an alternative to admission. In conclusion, telemonitoring at home was seen as an alternative to admission in a substantial proportion of medical ED patients. Facilitators for telemonitoring indicated by patients were

the Creative Commons Attribution License, which permits unrestricted use, distribution, and reproduction in any medium, provided the original author and source are credited.

**Data availability statement:** All relevant data are within the paper and its Supporting Information files.

**Funding:** The author(s) received no specific funding for this work.

**Competing interests:** The authors have declared that no competing interests exist.

guaranteed admission if telemonitoring failed and the possibility to contact the treatment team 24/7, while indicated barriers were related to disease severity and lack of someone to watch over the patient. Telemonitoring in acute care may serve as a potential alternative to admissions if facilitators are met.

## Author summary

This pilot study explored whether home telemonitoring could be a safe and acceptable alternative to hospital admission for patients seen in the emergency department (ED). Researchers surveyed 98 medical ED patients and consulted healthcare professionals to assess the feasibility of sending patients home with remote monitoring instead of admitting them to the hospital. Patients saw telemonitoring as a promising option if certain conditions were met. Key factors that made them more comfortable with the idea included guaranteed hospital admission if their condition worsened (96.9%), 24/7 access to their care team (90.8%), and having someone at home to assist them (72.4%). However, some patients were concerned about the severity of their illness, the need for treatments not possible at home, and safety. While 11.2% of patients believed admission could be avoided with telemonitoring, doctors and nurses were slightly more cautious. All patients who actually experienced telemonitoring found it easy to use and reassuring, and recommended this telemonitoring to family and friends. The study concludes that home telemonitoring could be a viable alternative to hospitalization for selected patients if safety and support are ensured. This approach may help reduce hospital crowding while maintaining patient comfort and care quality.

## Introduction

To relieve the increasing pressure on the healthcare system and to adjust to the preferences of patients [1], alternative forms of care are being explored. One of these alternatives is telemonitoring at home. By monitoring specific aspects of health and disease, patients can manage their health or disease within their own environment in close collaboration with and supervision by their treatment team. Continuous telemonitoring of vital signs has proven practical and is well-received, safe, and effective in decreasing care demands for a number of chronic conditions [2–5]. Whether these findings in chronic conditions can be extrapolated to acute conditions is, however, currently unclear. The acute care setting differs from the chronic setting, and patients who visit the emergency department (ED) are likely to respond differently than patients with chronic conditions. Adapting to new instruments and a new form of care is likely to be challenging.

To date, experience with telemonitoring for acute conditions in acute settings is scarce. Most experience stems from the COVID-19 pandemic [6–9]. As an alternative to admission, patients were sent home with oxygen, medication and instruments

to measure vital signs. Telemonitoring at home was shown to be feasible and safe, and many of the COVID-19 patients could be treated outside the hospital. However, there is limited experience with other acute conditions [10–14]. The results of these studies were promising, and the patients recovered faster and moved more at home than patients who were admitted [13].

This limited experience with home monitoring in acute conditions warrants more studies. Before home monitoring can be implemented on a larger scale, however, it is important to further investigate patient preferences regarding and experiences with home monitoring devices in the acute care setting. Additionally, methods should be developed to identify patients who are both suitable for and capable of being monitored at home [10,15,16]. If more information on preferences/experiences and methods for the identification of suitable candidates for homemonitoring becomes available, steps can be taken to implement a new form of acute care starting in the ED.

We therefore conducted a prospective pilot study to investigate the experiences of medical patients and acute care physicians with continuous telemonitoring started in the ED. We also investigated the preferences of ED patients regarding telemonitoring as continued care starting in the ED. In addition, we aimed to estimate the proportion of patients who could be treated at home instead of in the hospital by using telemonitoring.

## Methods

### Design/setting

This prospective pilot study was performed in the ED of Maastricht University Medical Centre, the Netherlands, which provides secondary and tertiary care. In the ED, about 7000 patients are assessed and treated by (acute) internists and their residents per year. On average, two thirds of these ED patients are admitted to the hospital.

In the study period, ED patients and their physicians and nurses were asked to complete questionnaires on home monitoring, and a subgroup of patients was then asked to try a telemonitoring system for at least 24 hours to test telemonitoring as a possible form of continued care.

### Patients

All adult (18+) ED patients who were assessed and treated by internists during 10 consecutive weekdays between 7:30 and 15:30h, were asked for informed consent and, if provided, included in the study. Exclusion criteria were: refusal of consent, being too severely ill or confused to answer the questions in a reliable way without putting too much strain on both patient and family (no consent possible), ED revisit (patients could be included only once during the study period), and a language barrier. All patients who provided consent were asked to complete a questionnaire (ED cohort).

Part of the ED patients were selected to form a second cohort (telemonitor cohort). Patients were eligible for this cohort if they provided additional informed consent to wear a telemonitoring system for at least 24 hours and to answer questionnaires on their experience with the telemonitor system. In addition, they had to live within a 15 km radius of the hospital. Furthermore, they were either about to be discharged home (i.e., judged not in need for admission) or were admitted to the hospital without requiring continuous monitoring of vital signs (i.e., not in need of continuous monitoring of vital signs or admission to either a medium or intensive care unit (MCU/ICU, resp.). We made this conservative selection of patients because assessment of safety was not within the scope of this study. The patients who were discharged home from the ED with the telemonitoring device were asked to measure their vital signs (blood pressure, heart rate, oxygen saturation and temperature) three times a day.

### Telemonitoring system

For this study, a vital sign monitoring device (Sensium, The Surgical Company, Abingdon, UK) was utilized [17]. The system consists of a wearable patch applied to the chest, and a lead guided through the axilla of the non-dominant arm. The

system connects wirelessly to either hospital-based computers or via cellular connection to a mobile app to reveal vital signs – i.e., respiratory rate, heart rate, axilla temperature, and changes over time. Thresholds for vital signs, as well as the duration over which these thresholds had to be exceeded before an alarm was triggered, were determined in advance. Once applied to the patient, the system required no patient action on their part, transmitting data autonomously.

Neither patients nor hospital staff were aware of the results of the measurements. All patients received care as usual with the exception of daily phone calls on their experiences. If the vital signs or alerts prompted further investigation or action, two of the investigators (experienced acute internists) contacted the treatment team (admitted patients) or patients (at home).

## Data collection and procedures

For the ED cohort, a *preference questionnaire* was used (S1 Text, in part based on [18,19]), and completed by all participants at the end of their ED visit. The questionnaire aimed to collect data on the availability of and skills regarding communication, medical and digital instruments (e.g., blood pressure monitor and digital skills [19]). In addition, preferences (facilitators and barriers) were evaluated regarding the prerequisites to be fulfilled to return home. Furthermore, the judgement of their ability to return home in their current condition was assessed, and, when applicable, on the reasons why returning home was not possible.

For the ED cohort, physicians and nurses completed a short *assessment questionnaire* on the potential of home monitoring as an alternative to admission, and on the capabilities of patients to use home monitoring in three domains: physical, cognitive, home support (S2 Text).

For the telemonitor cohort, an *experience questionnaire* regarding patient experience with telemonitoring was completed at the end of their monitoring period (S3 Text), in part based on the SUS questionnaire [20]. For this cohort, data generated by the telemonitor system were collected, and generated alarms were compared with either vital sign measurements during admission (measured as usual by nurses) or by self-measurements of the vital signs three times a day or vice versa (aberrant vital signs that were manually measured were compared with the monitor data).

The experience with the telemonitoring system of the two acute internists was evaluated at the end of the pilot using a short *experience questionnaire* (S4 Text).

Electronic medical charts were used to collect data on age, sex, means of transportation to the ED, triage category (using MTS: low (blue, green) or highly urgent (yellow/orange/red) ([21]), vital signs (necessary to calculate Modified Early Warning Score (MEWS) scores [22]), ED diagnosis (categorized according ICD-10 [23]), treatment started at the ED (oxygen yes/no, infusion yes/no, intravenous antibiotics yes/no), ED discharge destination (general ward, MCU/ICU, home), and length of hospital stay.

## Outcome measures

### Primary

- The proportion of patients satisfied with telemonitoring (telemonitor cohort).

### Secondary

- Proportion of patients able to return home from the ED with telemonitoring as an alternative to admission (ED cohort).
- Agreement between patient, physician and nurse assessments of being "fully capable" (in three domains: physical, cognitive and home support) for telemonitoring and for avoiding admission (ED cohort).

## Analysis and statistics

The main part of the analyses regarding the experience with telemonitoring and patient preferences were merely descriptive; for patient characteristics, preferences, experiences, and generated alarms, means (with SD), medians (with IQR) or proportions were calculated. When appropriate, comparisons were made using Student's-T, or Mann-Whitney U- or Chi-square tests. In case of missing data, valid percentages were calculated.

 

Agreement on judgment of being fully capable of returning home with telemonitoring between patients, and physicians or nurses was analyzed by calculating Cohen's kappa, whereas a kappa of 0-0.20 indicates slight, 0.21-0.40 fair, 0.41-0.60 moderate, 0.61-0.80 substantial, and >0.81 almost perfect agreement [24]. Regarding the agreement on possible avoidance of admission by using telemonitoring, Cohen's kappa was calculated as well (only calculated in patients who were admitted to the hospital).

We used IBM SPSS Statistics for Windows, version 28.0 (IBM Corp., Armonk, N.Y., USA), for the statistical analyses. P values <0.05 were considered statistically significant.

## Ethics

This study was approved by the Medical Ethics Committee of MUMC+ (METC approval: 2023–0248). Written informed consent was provided by all patients.

## Results

### Study population

In the study period, a total of 119 patients presented to the ED and were assessed and treated by internists (Fig 1). Of these, 21 were excluded. In total, 98 patients answered the questionnaire and formed the ED cohort, while 21 of these patients agreed to try the telemonitor system and formed the Telemonitor cohort. Of these, 12 returned home and 9 were admitted to the hospital (more details on their diagnoses in S1 Table).

The median age of the 98 patients was 68.5 (IQR: 53-78.3; Table 1). All but one were community-dwelling, and 66.0% were living with a roommate. The majority of patients were digitally skilled (advanced: 64.3%). In total, 54 (55.1%) patients were admitted to the hospital. Almost all patients had a mobile phone, and a slight majority had medical instruments and

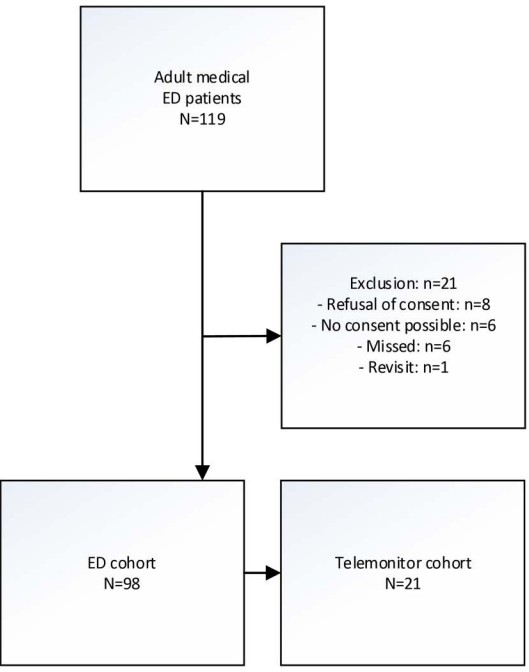

**Fig 1. Flowchart.**

PLOS Digital Health

**Table 1. Baseline characteristics.**

| | ED cohort N=98<br>N(%) or median [IQR] | Telemonitor cohort N=21<br>N(%) or median [IQR] |
|---|---|---|
| **Demographics** | | |
| Age (y) | 68.5 [53-78.3] | 70 [60.0-76.5] |
| ≥75 years | 35 (35.7) | 8 (38.1) |
| Male | 49 (50.0) | 12 (57.1) |
| **Living situation** | | |
| Community dwelling | 97 (99.0) | 20 (95.2) |
| Roommate* | 64 (66.0) | 14 (70.0) |
| Informal caretaker* | 82 (84.5) | 18 (90.0) |
| **Digital skills** | | |
| Not skilled-beginner | 28 (28.6) | 6 (28.6) |
| Moderately skilled | 7 (7.1) | 2 (9.5) |
| Advanced | 63 (64.3) | 13 (61.9) |
| **Acute care data/disease severity** | | |
| Transportation by ambulance | 17 (17.3) | 4 (19.0) |
| Triage code (MTS) | | |
| Blue/green | 38 (38.8) | 11 (52.4) |
| Yellow/orange/red | 60 (61.2) | 10 (47.6) |
| MEWS | 1 [0-9] | 2 [0-5] |
| MEWS≥3 | 28 (28.6) | 8 (38.1) |
| **Subspecialty** | | |
| General internal medicine | 52 (53.1) | 17 (81.0) |
| Gastroenterology | 19 (19.4) | 0 |
| Oncology | 14 (14.3) | 0 |
| Hematology | 4 (4.1) | 1 (4.8) |
| Immunology | 4 (4.1) | 1 (4.8) |
| Nephrology | 3 (3.1) | 1 (4.8) |
| Other | 2 (2.0) | 1 (4.8) |
| **ED diagnosis/ICD-10** | | |
| Infectious diseases | 36 (36.7) | 7 (33.3) |
| Digestive system | 17 (17.3) | 0 |
| Cardiovascular system | 14 (14.3) | 7 (33.3) |
| Blood/blood forming organs | 12 (12.2) | 2 (9.5) |
| Genitourinary system | 6 (6.1) | 1 (4.8) |
| Neoplasm | 4 (4.1) | 0 |
| Endocrine/metabolic | 3 (3.1) | 1 (4.8) |
| Other | 6 (6.1) | 3 (14.3) |
| **Continued care** | | |
| Admission to hospital | 54 (55.1) | 9 (42.9) |
| Regular ward | 51 (52.0) | 9 (42.9) |
| Medium care unit | 3 (3.1) | 0 |
| Length of hospital-stay (days) | 4 [1-58] | 5 [1-58] |

*denominator is the number of community dwelling patients

were able to use these independently (S2 Table). Only a few (n = 7; 7.1%) patients had experience with telemonitoring; none in the setting of acute care.

There were no significant differences in age, sex, triage code nor in discharge destination, between included and non-included patients (S3 Table).

## Patient preferences

Most patients preferred the phone as a way of contacting the treatment team (Table 2). The three most frequently mentioned facilitators for telemonitoring were guaranteed admission if necessary (96.9%), possibility to contact the treatment team 24/7 (90.8%) and the presence of someone to watch over/assist the patient (72.4%). In total, 51 (52.0%) patients indicated that they were unable to go home considering their current condition. The main reasons were the judgement that the necessary treatment could not be provided at home, feeling too severely ill, and judging it unsafe to go home.

## Home monitoring as an alternative to admission

Of 98 patients, 11 (11.2%) indicated that an admission could be avoided by using telemonitoring, while another 6 (6.1%) thought that this might be possible (Table 3). For professionals, these proportions were judged somewhat lower.

Agreement between patients and either physicians or nurses regarding avoidance of admission was slight-fair with kappa values of 0.240 and 0.210, respectively.

## Assessment of capability to use telemonitoring

In total, physicians judged 47 of 97 patients (48.5%) to be fully capable to use telemonitoring with regard to the three domains (physical, cognitive and home support; Table 4). The agreement with the patients' perspective was slight (kappa: 0.153). Agreement between patients nurses and was slight with a kappa of 0.161 (S4 Table).

**Table 2. Patient preferences.**

| | ED cohort (all patients) N = 98 N (%) | Telemonitor cohort N = 21 N (%) |
|---|---|---|
| **Preferred way of communication with professionals** | | |
| Phone | 74 (75.5) | 17 (81.0) |
| E-mail | 5 (5.1) | 0 |
| Video calling | 15 (15.3) | 3 (14.3) |
| App | 4 (4.1) | 1 (4.8) |
| **Facilitators for telemonitoring** | | |
| Guaranteed admission if necessary | 95 (96.9) | 21 (100.0) |
| Possibility to contact treatment team 24/7 | 89 (90.8) | 18 (85.7) |
| Someone to watch over me | 71 (72.4) | 12 (57.1) |
| Help with the measurements | 46 (46.9) | 9 (42.9) |
| Daily contact with the treatment team | 42 (42.9) | 9 (42.9) |
| Help in establishing contact with treatment team | 30 (30.6) | 5 (23.8) |
| **Barriers for telemonitoring (n = 51)** | | |
| Treatment not possible at home | 50 (98.0) | |
| Too severely ill | 42 (82.3) | |
| Unsafe to go home | 29 (56.9) | |
| Insufficient support at home | 13 (25.5) | |
| Not able to inform about my condition | 8 (15.7) | |
| Not able to establish contact | 5 (9.8) | |

**Table 3. Avoidance of admission by telemonitoring: three perspectives\*\*.**

|  | Patient N (%) | Physician N (%) | Nurse N (%) |
|---|---|---|---|
| Avoid admission: no | 51 (52.0) | 49 (50.5) | 47 (49.0) |
| Avoid admission: yes | 11 (11.2) | 7 (7.2) | 10 (10.4) |
| Avoid admission: maybe | 6 (6.1) | 6 (6.1) | 4 (4.2) |
| Home without telemonitoring | 30 (30.6) | 35 (36.1) | 35 (36.5) |
| *Missing** |  | 1 | 2 |

\*Valid percentages were calculated,

\*\*Agreement after exclusion of category "home without telemonitoring" (for patients who could go home, agreement on avoidance was not assessed). Kappa values: patient-physician: 0.240; patient-nurse: 0.210.

**Table 4. Agreement between patient and physician of ability to use telemonitoring.**

|  | Physician: Patient fully capable* | Physician: Patient not fully capable* |  |
|---|---|---|---|
| **Patient: fully capable*** | 10 | 37 | 47 |
| **Patient: not fully capable** | 33 | 17 | 50 |
|  | 47 | 54 | 97 |

kappa = 0.153; 1 missing

\*fully capable in all three domains (physical, cognitive, home support); not fully: not capable in at least one domain

## Experience with telemonitoring – patient perspective

In total, 21 patients used the telemonitor system (Table 1). Of these, 9 patients were admitted to the hospital, and 12 were discharged home. All patients reported to be satisfied or very satisfied with the instruction and explanation (insight), the easiness of use of the telemonitor system, and all but one (95.2%) were (very) self-confident in using the system (Fig 2A and 2B).

The concept of telemonitoring gave most patients (81.0%) a (very) reassuring feeling and none experienced problems with their privacy (Fig 2B). All patients indicated to be willing to use the system again, and would recommend it to their family or friends. All but three (85.7%) considered telemonitoring a good alternative to admission. In contrast, most patients (61.9%) indicated that they would not prefer to do the measurement their selves.

## Experience with telemonitoring - physician perspective

The telemonitoring system was easy to install at the ED, and patients could be quickly instructed within 15–20 minutes (Fig 3). It was easy to follow the patients from a distance, both the vital signs and technical alarms. It was however difficult to assist a patient from a distance when technical problems arose (e.g., detachment of the plaster or displacement of the temperature lead). In total, 90 alarms were triggered in 17 patients (median: 4 per patient; details in S5 Table). Part of these alarms could be explained by a misplaced temperature lead (n = 44).

## Discussion

In this prospective pilot study, we evaluated perspectives on and experience with telemonitoring in patients visiting the ED with an acute medical condition. In total, about 1 in 7 patients judged it possible/probable to return home with telemonitoring instead of being admitted. The judgement of physicians and nurses on being suitable of returning home instead of

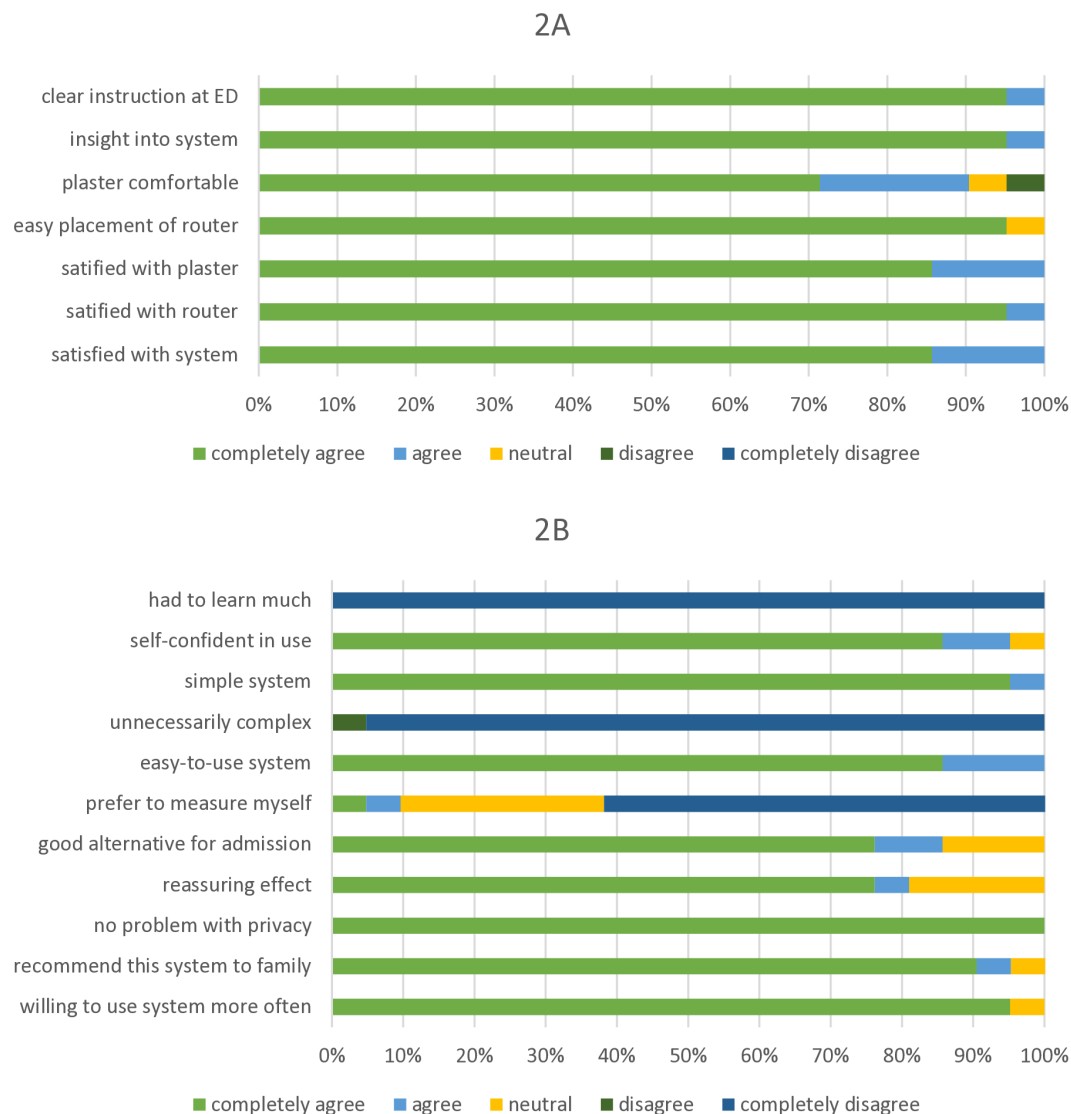

**Fig 2. A) Patient experience regarding instruction and set up. B) Patient experience with telemonitoring.**

being admitted was comparable, however, agreement on this judgement between patients and professionals was only slight-fair. The main reasons not being able to go home were, according to patients, being too seriously ill, the need for treatment that can only be provided in the hospital and the home situation not being safe. The three most mentioned facilitators were guaranteed admission, if necessary, the possibility to contact the treatment team 24/7, preferably by phone, and the presence of someone who could watch over the patient. A subset of these patients tried a telemonitoring system either at home or during admission. The system was feasible for installation in and suited in the workflow of the ED, and was very well-received (easy-to-use, privacy, willing to use it again).

Alternatives to admission in the hospital are currently developed to relieve the strain in the health care system and to provide chronic care that is both well-received by patients and safe. Experiences gained in chronic care can now be applied to acute care [4,25]. Successful implementation of telemonitoring in (crowded) EDs in patients who may not

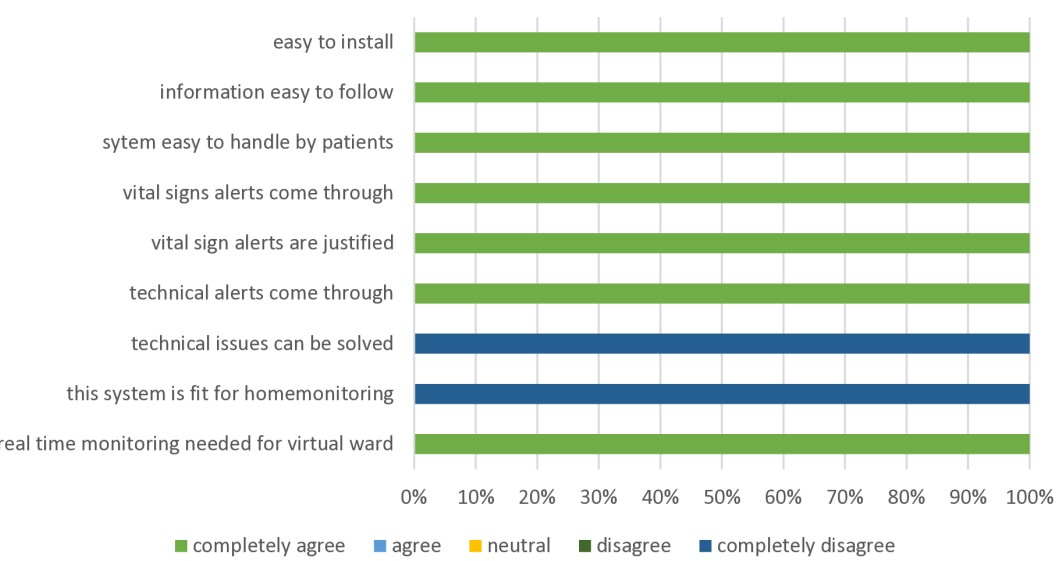

**Fig 3. Physician experience.**

be stable, is however challenging [16,26]. We believe that before we can start building a new concept, it is important to evaluate the perspectives of patients and acute care professionals. The current study reveals that, although most patients indicated to have no experience with home monitoring, many indicated to be open for this new form of acute care. The reasons for not willing to go home were to be expected, but some of these barriers may be solved in the future, for instance by organizing intravenous medication services. In addition, the positive experience of the telemonitor cohort shows that home monitoring was well-received, which is also seen in other studies [27,28]. The acute care professionals, as well, saw the potential of home monitoring in avoiding admissions. It is important to note that neither patients nor professionals had experience with home monitoring in acute care. Gaining more insight in the opportunities and barriers of telemonitoring in patients and care professionals can be evaluated in qualitative studies and in clinical trials, which involve patients in the design and interpretation of the results.

Several components for home monitoring for home monitoring as an alternative to admission in the hospital have been identified [15]: medical care, nursing care, remote monitoring, technology/instruments and care coordination. Regarding medical care, interestingly, almost all admitted patients received intravenous fluids/medication or oxygen. It is important to evaluate whether this treatment was necessary as not all medical care can easily and timely be applied in the home situation. It is possible that starting intravenous fluids at the moment of admission is just a routine procedure. Nursing care needs were not evaluated in this study. The presence of someone to watch over the patient was an important prerequisite for home monitoring, but this does not necessarily mean that that someone is a care professional. Interestingly, the patients had many medical instruments at home. This means that it is possible to use these instruments for home monitoring, and/or to build on the skills of the patients. Most of the patients in the telemonitor cohort, however, indicated that they preferred the continuous monitoring of vital signs instead of self-measurements. An additional argument for providing all monitoring material is that this material is validated, and that data can be sent to the treatment team and electronic patient charts. We did not perform analyses into the technology/instruments needed nor into care coordination.

To date, only a few, small randomized controlled studies, in diverse groups of patients, were performed on home monitoring in acute care [12–14]. These small studies showed that telemonitoring was cost-effective and safe. Interestingly, patients treated in their homes moved more than hospitalized patients.

It is further important to be able to distinguish between those in need for admission, and those fully fit for further treatment at home. The slight agreement between patients and professionals on this issue that we found is interesting, and probably, at least in part, results from a lack of communication about telemonitoring between patients and professionals, but this warrants further investigation. Risk stratification scores could be useful for making this distinction [13].

Evaluation of safety was not within the scope of our study because it was not possible to assess all generated alerts as no corresponding measurements were made in many cases. In addition, patients were, for safety reasons, selected on basis of the absence of need for continuous monitoring. For future implementation of telemonitoring in acute care, safety should be established, measurements should be robust, and the most important vital signs should be selected.

## Limitations

Limitations of the study are that we had to adapt questionnaires as no relevant validated questionnaires applied to this study [16,26]. More in-depth information on the perspectives of patients and professionals can be evaluated in qualitative studies, which may build upon the facilitators/barriers identified in this study. In addition, we performed the study during office-hours for practical reasons. This resulted in a relatively small study population, possibly resulting in a lack of power regarding analyses of, for instance, judgement of avoidance of admission. It is also possible that the proportion of patients willing and able to go home with home monitoring in evenings, nights and weekend differs from that during the office hours when this study was conducted. However, in a recent, similar study that included more patients, including those presenting in the evening, comparable proportions were found [18]. Another limitation is that patients nor professionals were familiar with the concept of telemonitoring at home. It is therefore possible that they underestimated the potential of telemonitoring. It would further be interesting to involve patients and care professionals other than physicians in the interpretation of the results of future studies that focus on preferences and experiences of new care concepts.

## Conclusion

Home monitoring was seen as an alternative to admission in a substantial proportion of medical ED patients. Barriers for home monitoring indicated by patients were related to disease severity, need for treatment considered to warrant admission and to a lack of someone to watch over the patient, while a guaranteed admission and 24/7 possibility to contact the treatment team were facilitators for home monitoring. Although patients had many medical instruments at home, most monitored patients preferred the continuous automated measurements. The agreement in assessment of being fully fit for home monitoring between patients and professionals was low. No conclusions on safety can be drawn, but the telemonitor system tested was feasible for use in the workflow of the ED, well-received and perceived as comfortable, and the concept seems to have potential in reducing the pressure on the health care system.

## Supporting information

**S1 Text. Preference questionnaire (ED cohort).**
(DOCX)

**S2 Text. Assessment questionnaire for nurses and physicians.**
(DOCX)

**S3 Text. Experience questionnaire patients (telemonitor cohort).**
(DOCX)

**S4 Text. Experience questionnaire investigators.**
(DOCX)

**S1 Table. ED main diagnoses telemonitorcohort.**
(DOCX)

**S2 Table. Availability of and proficiency with devices and instruments, and digital skills.**
(DOCX)

**S3 Table. Inclusion bias analysis.**
(DOCX)

**S4 Table. Agreement between patient and nurse on assessment of patient capability to use telemonitoring.**
(DOCX)

**S5 Table. Alarms in telemonitorcohort.**
(DOCX)

**S1 Data. Dataset.**
(SAV)

## Author contributions

**Conceptualization:** Patricia M Stassen.

**Data curation:** Patricia M Stassen.

**Formal analysis:** Noortje Zelis, Patricia M Stassen.

**Investigation:** Dewa Westerman, Anouk Schevers, Patricia M Stassen.

**Methodology:** Noortje Zelis, Dewa Westerman, Anouk Schevers, Nicole v Eldik, Patricia M Stassen.

**Resources:** Nicole v Eldik.

**Supervision:** Patricia M Stassen.

**Writing – original draft:** Noortje Zelis, Dewa Westerman, Patricia M Stassen.

**Writing – review & editing:** Noortje Zelis, Dewa Westerman, Anouk Schevers, Nicole v Eldik, Patricia M Stassen.

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
