## [Decision Letter · Decision Letter 0]

Response to Reviewers
Revised Manuscript with Track Changes
Manuscript
**Journal Requirements:**
**Additional Editor Comments (if provided):**
**Reviewers' Comments:**

**Comments to the Author**

1. Does this manuscript meet PLOS Digital Health’s publication criteria?

Reviewer #1: Partly

Reviewer #2: Yes

2. Has the statistical analysis been performed appropriately and rigorously?

Reviewer #1: Yes

Reviewer #2: Yes

3. Have the authors made all data underlying the findings in their manuscript fully available (please refer to the Data Availability Statement at the start of the manuscript PDF file)?

Reviewer #1: Yes

Reviewer #2: Yes

4. Is the manuscript presented in an intelligible fashion and written in standard English?

Reviewer #1: No

Reviewer #2: Yes

Reviewer #1: Dear authors,

The aim of the study is good. It needs to be supported in the introduction. The introduction part is simple and

careless. Also, some undisclosed data is quite important and necessary. I uploaded the file.

The arrangements are presented on the file.

Reviewer #2: This is a very interesting and timely paper given the exponential increase in home monitoring technologies being implemented across the globe. While the authors recognize the lack of research in this area, they overstate the safety aspects of this pilot. The potential application of the findings is quite limited given the small sample size and limited information gathered in this study. It still remains unclear whether telemonitoring should be used as an alternative for ED admission.

Of most concern is the fact that the telemonitoring cohort that returned to hospital received more initial treatment than the ED cohort. (See Table 1, Initial treatment after admission, page 10). This sees to indicate that they were much sicker and required more care and attention than the ED cohort. This needs to be more clearly addressed as this program may not in fact be as useful a strategy as the authors may have thought.

In the title they say the study focuses on patient perspectives and first experiences. However there is a substantial lack of qualitative data to describe in-depth these perspectives and experiences. The authors are encouraged to rephrase the title and add their methodology for clarity.

The inclusion exclusion criteria are unclear. For example – what does ‘severely ill’ mean? Most individuals receiving care at the ED could be considered ‘severely ill’. So more clarity is needed.

The rationale for the groups was unclear. (See lines 103-110, page 4, the second paragraph under patients). Please add more detail.

On the tables, please specify N and % at the top of the specific columns. For example, as Table 1 reads now, column 1is noted to show N (%) while this is actually the case for columns 2 and 3. Column 1 actually shows the Patient characteristic.

Table 4 requires more detail as it simply states Patient: capable. Would be helpful to specify capable of what? The narrative notes three domains. In table 3 must they be capable of all three domains, or one or more of the three? It is hard to follow the details of what was done here.

Bar charts are hard to read when printed in black and white. Suggest the authors recolour these using black and white patterns with outlined bars to ensure readers can follow. Also, capitalization is needed in the headings on Figures 2A, 2B, and 3.

Further details on the specifics of the monitoring are needed. How did participants learn how to use them. Were they self-monitored or monitored by a care partner? Was a specific type of technology used? If so, please specify this. Not enough information is available for a reader who may be interested to reproduce or expand on this work.

**Do you want your identity to be public for this peer review?** For information about this choice, including consent withdrawal, please see our Privacy Policy

Reviewer #1: No

Reviewer #2: No

**Figure resubmission:****Reproducibility:** To enhance the reproducibility of your results, we recommend that authors of applicable studies deposit laboratory protocols in protocols.io, where a protocol can be assigned its own identifier (DOI) such that it can be cited independently in the future. Additionally, PLOS ONE offers an option to publish peer-reviewed clinical study protocols. Read more information on sharing protocols at https://plos.org/protocols?utm_medium=editorial-email&utm_source=authorletters&utm_campaign=protocols

---

## [Decision Letter · Decision Letter 1]

Response to Reviewers
Revised Manuscript with Track Changes
Manuscript
**Additional Editor Comments (if provided):**
**Reviewers' Comments:**

**Comments to the Author**

Reviewer #3: (No Response)

Reviewer #4: (No Response)

Reviewer #5: (No Response)

publication criteria?

Reviewer #3: Yes

Reviewer #4: Partly

Reviewer #5: Partly

3. Has the statistical analysis been performed appropriately and rigorously?

Reviewer #3: Yes

Reviewer #4: Yes

Reviewer #5: I don't know

4. Have the authors made all data underlying the findings in their manuscript fully available (please refer to the Data Availability Statement at the start of the manuscript PDF file)?

Reviewer #3: Yes

Reviewer #4: Yes

Reviewer #5: Yes

5. Is the manuscript presented in an intelligible fashion and written in standard English?

Reviewer #3: Yes

Reviewer #4: No

Reviewer #5: Yes

Reviewer #3: This is a timely and well-motivated pilot study exploring an emerging area—telemonitoring in acute care. The revised manuscript has addressed many of the initial reviewer concerns thoughtfully and effectively. It is significantly improved in framing its objectives, clarifying methods, and reinforcing its identity as a feasibility and experience-based study rather than a safety evaluation. However, several areas would still benefit from further clarification and precision to strengthen reproducibility and overall interpretability:

• While the authors now clarify that safety was not assessed, statements like “telemonitoring can be used in acute care as alternative for admissions if prerequisites are met and barriers removed” might still feel conclusive. Suggest rephrasing as “telemonitoring may serve as a potential alternative…” in the abstract and conclusion to avoid overinterpretation.

• Reviewer #2’s concern about the telemonitoring setup remains only partially addressed. Key details—such as the specific type, name, or model of the device used (if applicable)—are still lacking. This information is important to ensure reproducibility and allow others to build on this work.

• The concept of being “capable” (Table 4) is still not fully clear. Since this refers to three domains (physical, cognitive, support), it would be helpful to rephrase as “fully capable in all three domains with an asterisk” consistently across tables and text. The asterisk can be used explain the 3 domains in the table’s caption.

• Table 1 reports separately on patients being community-dwelling, with a roommate, and having an informal caregiver. However, the relationship between these categories remains unclear. Is there overlap between those who live with a roommate and those who have an informal caregiver? This is important because, if I interpret “roommate” and “informal caregiver” as distinct subsets of the community-dwelling group, the reported percentages do not logically add up. Additionally, it appears that the one non–community-dwelling patient was omitted from Table 1—please clarify whether this was intentional.

• Table 3, clearer legends are needed to explain how “Home without telemonitoring” fits within the other categories (i.e., are they excluded from kappa analysis?).

• The reviewer 2 comments on Table 4 was not fully addressed. Clarify that physicians rated capability based on the combined three domains, and also, specify whether being classified as “not fully capable” means the patient failed in any one of the domains, or in all three.

• It is mentioned that 12 patients went home with the telemonitoring device. It would be beneficial to clarify if their experiences (as compared to admitted patients) were different in terms of satisfaction or usability.

• The results section is very data-heavy. Consider summarising main takeaways more narratively in paragraph form for better readability (especially around patient vs provider agreement).

• I recommend making the limitations section a distinct subheading under the Discussion. It would strengthen the manuscript to explicitly state that this was a convenience sample collected during office hours, which may affect generalisability to the full ED population, including evenings and weekends.

• MCU/ICU explained well in text but define MEWS in full at first mention for clarity.

• In the Methods (Lines 85–86), please state the hospital’s location and country explicitly (e.g., “Maastricht University Medical Centre, Netherlands”). This provides helpful context for international readers.

Reviewer #4: Thank you for the opportunity to review this study, "Telemonitoring at home for provision of continued care in the acute care setting. A prospective pilot study focusing on first experiences of patients". This pilot prospective study explored ED patients' and healthcare professionals' perspectives on home telemonitoring as an alternative to hospital admission. Eligible patients (adult, assessed by internists, and without exclusion criteria such as severe illness, confusion, or language barriers) completed questionnaires about their willingness and preferences for telemonitoring, and a subset was enrolled in an actual telemonitoring program if they lived within 15km of the hospital and were either discharged or admitted without need for continuous hospital monitoring.

While the topic of this study and the actual methodology and analysis seemed to be relevant and useful, the paper requires significant revision. Much of the need for review may stem from how language was used throughout the article, which can benefit from a thorough review by an English language expert. Additionally, the abstract needs to be reviewed thoroughly to include information that is later mentioned in the article, but missing from the abstract (e.g. the 3 questions being investigated in the research). Please see the following for detailed feedback.

In the abstract it's not clear what is meant by the percentages in the Abstract's Results section. Are authors giving percentages of patients who met the prerequisites for telemonitoring? If so, please clarify accordingly: "The prerequisites for telemonitoring were guaranteed admission if necessary (96.9%), possibility to contact the treatment team 24/7 (90.8%) and presence of someone to watch over the patient (72.4%)."

"Main barriers were: treatment could not be provided at home, feeling too ill, and judging it unsafe to return home"  Barriers to what? Do authors mean barriers to being recruited for telemonitoring as part of the study (since a subset was started on telemonitoring)? Barriers that were identified by patients or doctors in a questionnaire?

"In total, 11.2% of patients indicated that admission could be avoided using telemonitoring, while 6.1% thought this might be possible."  If patients indicated that admission could be avoided did they also not think this might be possible? The distinction between these two groups of patients is unclear.

"agreement with patients was slight-fair"  please indicate in the Methods section of the abstract how agreement was statistically ascertained.

"Prerequisites were guaranteed admission and 24/7 contact with the treatment team..."  Again, in the conclusion section of the Abstract, it's unclear what authors mean by "prerequisites". Do they mean "facilitators" as opposed to "barriers"?

"...barriers were related to disease severity and lack of someone to watch over the patient. Telemonitoring can be used in acute care as alternative for admissions if prerequisites are met and barriers removed."  What do authors mean by "barriers removed"? How does one remove the barrier of disease severity? Do they mean if exclusion criteria are met?

52 In order to relieve the increasing pressure on the healthcare system and to adjust to

53 preferences of patients (1), alternative forms of care are being explored.

 please actually give the magnitude of the pressure you are referencing (increasing ED usage over the years, worsening ED wait times, etc); please mention actual prefrences of patients so readers can know it aligns with the solution being presented here.

PLEASE REVIEW LANGUAGE / GRAMMAR:

- ABSTRACT: In this prospective pilot study, we asked medical ED patients on their perspectives on home monitoring. In a subset, we started and evaluated telemonitoring.

- 63 To date, experience with telemonitoring in acute conditions in acute settings is scarce.  ...telemonitoring of acute conditions...

- 64 As alternative for admission,  As an alternative to admission

- 66 Telemonitoring at home turned out feasible and safe... Telemonitoring at home was shown to be feasible and safe; Please give citation.

- The sentences in lines 66-69 seem to be jumbled. Currently:

66 Telemonitoring at home turned out feasible and safe, and many patients could be treated

67 outside the hospital. However, there is limited experience with other acute conditions (10-14)

68 The results were promising, and the patients recovered faster and moved more at home than

69 patients who were admitted (13).

Did authors mean to say: Telemonitoring at home turned out feasible and safe, and many patients could be treated outside the hospital. The results were promising, and the patients recovered faster and moved more at home than patients who were admitted (13). However, there is limited experience with other acute conditions (10-14)

- Unclear what authors are trying to say:

71 Before home monitoring can be implemented on a larger scale, however, patient experience

72 with home monitoring devices and their preferences should be further explored in the acute

73 care setting, as are the ways to select patients who are suitable for and capable of using home

74 monitoring (10, 15, 16)

Are they trying to say: Before home monitoring can be implemented on a larger scale, however, patient experience with home monitoring devices and their preferences should be further explored in the acute care setting, as [SHOULD BE METHODS] to select patients who are suitable for and capable of using home monitoring (10, 15, 16)

Please revise the wording here ("more information on these aspects"? This is a strange way to word what they might be trying to say)

74 If more information on these aspects becomes available, the

75 necessary steps can be made to implement a new form of acute care starting in the ED

What does "aimed to retrieve the proportion of patients..." mean? Do you mean "estimate the proportion of patients..."?

79 In addition, we aimed to retrieve the

80 proportion of patients who could be treated at home instead of in the hospital by using

81 telemonitoring

In the introduction section of the article (lines 66-81) the authors identify 3 parts to their investigation:

(1) investigate the preferences of ED patients regarding telemonitoring as continued care starting in the ED

(2) investigate the experiences of medical patients and acute care physicians with continuous telemonitoring started in the setting of an ED

(3) estimate "proportion of patients who could be treated at home instead of in the hospital by using telemonitoring"

The third goal of this paper is not mentioned in the abstract's methods.

109 about to be sent home  about to be discharged home from the ED (not "about to be sent home")

What do you mean "aimed to retrieve"? "Telemonitoring concept"? Please revise this sentence:

112 because we aimed to retrieve

113 experiences of patients with the telemonitoring concept;

What do you mean in the following sentence? Do you mean "the telemonitor cohort were asked to measure..."? Or do you mean that there are a group of telemonitor patients who did not return home such that you specified in the sentence "telemonitor patients who returned home"?

114 The telemonitor patients who returned home were asked to measure

115 their vital signs (blood pressure, heartrate, oxygen saturation and temperature) three times a

116 day

In the Methods section:

Unclear whether patients who returned home had actions that were required (i.e. asked to measure their vital signs as mentioned in lines 114 and 115 OR no actions were required from the patients as mentioned in lines 124 and 125):

114 The telemonitor patients who returned home were asked to measure

115 their vital signs (blood pressure, heartrate, oxygen saturation and temperature) three times a

116 day.

VS

119 For this study, a telemonitoring device consisting of a plaster, applied to the chest, and a lead

120 guided through the axilla of the non-dominant arm. The plaster and lead continuously

121 measure the respiratory rate, heartrate, axilla temperature, and generate an output every 2

122 minutes. The thresholds of the vital signs and of the period during which the thresholds

123 before an alarm was raised were agreed in advance. The data were transmitted by a portable

124 router, which in turn transmitted the measurements and alarms via a web application. No

125 actions were required from the patients

From lines 119-125 it seems telemonitoring was happening in two places: at home for patients who were discharged from the ED; and in the hospital for patients who did not require continuous monitoring. Perhaps, revise the title, which indicates telemonitoring is happening at home: Telemonitoring at home for provision of continued care in the acute care setting. A prospective pilot study focusing on first experiences of patients

"...before an alarm was raised..."? Language needs to be revised to suit an academic journal.

122 The thresholds of the vital signs and of the period during which the thresholds

123 before an alarm was raised were agreed in advance.

Do you mean: "The thresholds for the vital signs, as well as the duration over which these thresholds had to be exceeded before an alarm was triggered, were determined in advance."

Similarly on line 299: "In total, 90 alarms were raised in 17 patients..."  For an academic journal, alarms are "triggered"...not "raised".

Don't use "filled in" to refer to "completed":

133 on (17, 18)), and filled in by all participants... - COMPLETED by all participants

140 For the ED cohort, physicians and nurses filled out a short assessment questionnaire...  physicians and nurses COMPLETED a short assessment questionnaire...

Throughout the article it seems the authors use the term "retrieve" quite frequently and incorrectly to indicate "collect" or "evaluate". They are using "retrieve" in a way that is atypical in the English language ("retrieve" is typically used to refer to fetching something that has been archived, not to refer to the collecting of or evaluate of data). Please revise this usage:

134 aimed to retrieve data on the availability of and skills regarding communication, medical and

135 digital instruments (e.g. blood pressure monitor and digital skills (18)). In addition,

136 preferences were retrieved...

152 The experience with the telemonitoring system of the two acute internists was retrieved at the...

311 In this prospective pilot study, we retrieved perspectives...

330 ...it is important to retrieve the perspectives...

340 retrieved in qualitative studies...

345 It is important to retrieve whether this

348 Nursing care needs were not retrieved in this study. T

Unsure what authors mean by "questions were asked on the judgement of their ability...":

137 Furthermore, questions were asked on the judgement of their ability to return home in their

138 current condition,

Authors can use the term "retrieve" on line 155 (I think they may be getting "retrieve" and "collect" confused:

155 Electronic medical charts were used to collect data on age, sex, means of transportation to the

156 ED  Electronic medical charts were used to retrieve data on age, sex, means of transportation to the ED...

Please revise this sentence "between by patient" doesn't make sense:

168 Agreement between by patient, physician and nurse assessments of being “fully capable”

169 (physical, cognitive and home support) for telemonitoring and for avoiding admission.

It's not clear which, if any, of the outcomes listed in the primary and / or secondary outcomes list include the questionnaire data about patient perspectives on home monitoring.

162 Outcome measures

163 Primary

164 The proportion of patients satisfied with telemonitoring (telemonitor cohort)

165 Secondary

166 Proportion of patients willing and able to return home from the ED with telemonitoring as

167 alternative for admission

168 Agreement between by patient, physician and nurse assessments of being “fully capable”

169 (physical, cognitive and home support) for telemonitoring and for avoiding admission.

170 The number and nature of generated alarms.

Please fix the Table 1 so the "n" for the row about initial treatment after admission lines up with the columns; the way it's represented now is not clear which n belongs where: Initial treatment after admission (n=54;

n=9)

Unclear how the authors are using the term "prerequisites". In some places it seems the authors are using the term "prerequisite" to refer to "preferences". In other places it seems they use the term "prerequisite" to refer to "patient stated requirement for telemonitoring" (which can be "patient prerequisite for telemonitoring"). Please clarify.

229 The

230 three most frequently mentioned prerequisites for telemonitoring were guaranteed admission

231 if necessary (96.9%), possibility to contact the treatment team 24/7 (90.8%) and the presence

232 of someone to watch over/assist the patient (72.4%).

Table 2: Prerequisites for telemonitoring

271 In total, 21 patients used the telemonitor system. Their details are depicted in Table 1.  Just cite the table: In total, 21 patients used the telemonitor system (Table 1).

For figures 2 and 3, please include 95% confidence intervals.

340 ...which involve patients in the setup...  which involve patients in STUDY DESIGN

359 These studies were positive – in terms of costs and safety.  What do you mean "positive"? Please just state the effect: e.g. Studies have demonstrated safety and cost-effectiveness...

Do not need the second comma

359 Interestingly,

360 patients treated in their homes, moved more than hospitalized patients. - Interestingly, patients treated in their homes moved more than hospitalized patients.

Reviewer #5: Title: Telemonitoring at home for provision of continued care in the acute care

setting. A prospective pilot study focusing on first experiences of patients

Comments:

1. The original submission focused on the use of telemonitoring as an alternative to hospital admission and emergency room admissions. In response to the first submission reviewer's comments the authors have now changed the title to one which frankly is less clear that of telemonitoring as a prelude for provision of subsequent acute care. It would be best to stick to what was originally decided as the focus of the study rather than post hoc shifting titles to a review or comment which then is really not aimed at what the study was all about. I suggest the authors think this through carefully and come up with a crisp title

2. Similarly, the same follows through in the abstract. The abstract states that telemonitoring may be used to reduce admissions to the ER and that they performed the pilot study to gain the perspectives and experiences of patients with telemonitoring - yet that sentence doesn't have a completion. Did the study aim at looking at reduction of ED admissions, or the experience in the ED or subsequent care??? The second sentence in the introduction of abstract needs to be fixed.

3. In the Results section of the abstract. This sentence needs to be clarified. “For professionals, these proportions were lower, and agreement with patients was slight-fair. ????

4. In the conclusion of the abstract. These sentences need to be clarified. “Prerequisites were guaranteed admission and 24/7 contact with the treatment team, while barriers were related to disease severity and lack of someone to watch over the patient. Telemonitoring can be used in acute care as alternative for admissions if prerequisites are met and. barriers removed. “ Barriers ??? – not clear

5. Entry criteria are still not clear. Was it all commers?

6. Need a breakdown of what were the disease – more granularity that what was provided , e.g. for CV there may be differences for a patient with HTN or chronic angina vs one with stuttering angina or unstable angina – need this kind of information ( or if unavailable need to comment on tis in the discussion) , the acute conditions being monitored, and/or Diagnosis at the time of admission. Very important.

7. Discussion. Line 349. Several building blocks for home monitoring as alternative for admission in the hospital have been identified (15): Building blocks. – not clear, poor choice of term

8. Discussion Line 405. The agreement in assessment of being fully fit for home

monitoring between patients and professionals was low. The numbers were so low. Need t comment on that. Its hard to distinguish patient vs professional opinion.

Overall, this is an important study, but the message is getting lost with lack of clarity. Suggest carefully outline the top 3-5 bullet points that you gleaned from the study – put these in the abstract and in the discussion and the paper will be much better.

**Do you want your identity to be public for this peer review?** For information about this choice, including consent withdrawal, please see our Privacy Policy

Reviewer #3: No

Reviewer #4: No

Reviewer #5: No

**Figure resubmission:****Reproducibility:** To enhance the reproducibility of your results, we recommend that authors of applicable studies deposit laboratory protocols in protocols.io, where a protocol can be assigned its own identifier (DOI) such that it can be cited independently in the future. Additionally, PLOS ONE offers an option to publish peer-reviewed clinical study protocols. Read more information on sharing protocols at https://plos.org/protocols?utm_medium=editorial-email&utm_source=authorletters&utm_campaign=protocols

---

## [Decision Letter · Decision Letter 2]

Response to Reviewers
Revised Manuscript with Track Changes
Manuscript
**Additional Editor Comments (if provided):**
**Reviewers' Comments:**

**Comments to the Author**

Reviewer #4: All comments have been addressed

Reviewer #5: All comments have been addressed

publication criteria?

Reviewer #4: Yes

Reviewer #5: Yes

3. Has the statistical analysis been performed appropriately and rigorously?

Reviewer #4: Yes

Reviewer #5: Yes

4. Have the authors made all data underlying the findings in their manuscript fully available (please refer to the Data Availability Statement at the start of the manuscript PDF file)?

Reviewer #4: Yes

Reviewer #5: Yes

5. Is the manuscript presented in an intelligible fashion and written in standard English?

Reviewer #4: Yes

Reviewer #5: Yes

Reviewer #4: Thank you for attempting to address my previous comments. Please conduct an English language review with someone who is a native speaker of the English language and who is not one of the co-authors. Once this is done I agree the manuscript is ready for publication. But please do the English language review.

Reviewer #5: Additional Suggested changes

Abstract

Line 49. All telemonitored patients were satisfied with the easiness of use and comfortability of the system, which gave most patients a reassuring feeling and was considered an alternative to admission.

Suggested edits. All telemonitored patients were satisfied with the easiness ease of use and comfortability comfort of the system, which gave most patients a reassuring feeling reassurance and was considered an alternative to admission

Methods

Line 135. For this study, a telemonitoring device consisting of a plaster, applied to the chest, and a lead guided through the axilla of the non-dominant arm (17). The plaster and lead continuously measure the respiratory rate, heartrate, axilla temperature, and generate an output every minutes. The thresholds for the vital signs, as well as the duration over which these thresholds had to be exceeded before an alarm was triggered, were determined in advance. The data were transmitted by a portable router, which in turn transmitted the measurements and alarms via a web application. No other actions were required from the patients to use this telemonitor system.

Suggested edits: For this study, a wireless vital sign monitoring device (Sensium®, The Surgical Company, Abingdon, UK) was utilized. The system consists of a wearable patch applied to the chest, and a lead guided through the axilla of the non-dominant arm (17). The patch connected wirelessly to either hospital-based computers or via cellular connection to a mobile app to reveal vital sign history – i.e. respiratory rate, heart rate, axilla temperature, and changes over time. Thresholds for vital signs, as well as the duration over which these thresholds had to be exceeded before an alarm was triggered, were determined in advance. Once applied to the patient the system required no patient action on their part, transmitting data autonomously.

Line 194: Student T tests, Chi2 tests or Mann Whitney U tests,

Suggested edits: Student’s t-test, Chi-squared test or Mann-Whiteny tests

**Do you want your identity to be public for this peer review?** For information about this choice, including consent withdrawal, please see our Privacy Policy

Reviewer #4: None

Reviewer #5: **Yes: ** Marvin J. Slepian

**Figure resubmission:****Reproducibility:** To enhance the reproducibility of your results, we recommend that authors of applicable studies deposit laboratory protocols in protocols.io, where a protocol can be assigned its own identifier (DOI) such that it can be cited independently in the future. Additionally, PLOS ONE offers an option to publish peer-reviewed clinical study protocols. Read more information on sharing protocols at https://plos.org/protocols?utm_medium=editorial-email&utm_source=authorletters&utm_campaign=protocols

---

## [Decision Letter · Decision Letter 3]

Telemonitoring starting in the emergency department as alternative to acute hospital admission.

A prospective pilot study focusing on patient preferences and first experience

PDIG-D-25-00071R3

Dear Dr. Stassen,

We are pleased to inform you that your manuscript 'Telemonitoring starting in the emergency department as alternative to acute hospital admission.

A prospective pilot study focusing on patient preferences and first experience' has been provisionally accepted for publication in PLOS Digital Health.

Best regards,

Haleh Ayatollahi

Section Editor

PLOS Digital Health

**Additional Editor Comments (if provided):**

**Reviewer Comments (if any, and for reference):**

Reviewer's Responses to Questions

**Comments to the Author**

Reviewer #4: All comments have been addressed

Reviewer #5: All comments have been addressed

publication criteria?

Reviewer #4: Yes

Reviewer #5: Yes

3. Has the statistical analysis been performed appropriately and rigorously?

Reviewer #4: Yes

Reviewer #5: Yes

4. Have the authors made all data underlying the findings in their manuscript fully available (please refer to the Data Availability Statement at the start of the manuscript PDF file)?

Reviewer #4: Yes

Reviewer #5: Yes

5. Is the manuscript presented in an intelligible fashion and written in standard English?

PLOS Digital Health does not copyedit accepted manuscripts, so the language in submitted articles must be clear, correct, and unambiguous. Any typographical or grammatical errors should be corrected at revision, so please note any specific errors here.

Reviewer #4: Yes

Reviewer #5: Yes

Reviewer #4: (No Response)

Reviewer #5: The authors have further improved the paper with the latest revision

**Do you want your identity to be public for this peer review?** For information about this choice, including consent withdrawal, please see our Privacy Policy

Reviewer #4: No

Reviewer #5: **Yes: ** Marvin J Slepian
